# Mitophagy: A Bridge Linking HMGB1 and Parkinson’s Disease Using Adult Zebrafish as a Model Organism

**DOI:** 10.3390/brainsci13071076

**Published:** 2023-07-16

**Authors:** Khairiah Razali, Mohd Hamzah Mohd Nasir, Jaya Kumar, Wael M. Y. Mohamed

**Affiliations:** 1Department of Basic Medical Sciences, Kulliyyah of Medicine, International Islamic University Malaysia (IIUM), Kuantan 25200, Pahang, Malaysia; kyaaahrzl@gmail.com; 2Department of Biotechnology, Kulliyyah of Science, International Islamic University Malaysia (IIUM), Kuantan 25200, Pahang, Malaysia; hamzahn@iium.edu.my; 3Department of Physiology, Faculty of Medicine, UKM Medical Centre, Kuala Lumpur 56000, Selangor, Malaysia; jayakumar@ukm.edu.my; 4Clinical Pharmacology Department, Menoufia Medical School, Menoufia University, Shebin El-Kom 32511, Menoufia, Egypt

**Keywords:** Parkinson’s disease, high-mobility group box 1, MPTP, mitochondrial dysfunction, neuroinflammation, zebrafish

## Abstract

High-mobility group box 1 (HMGB1) has been implicated as a key player in two critical factors of Parkinson’s disease (PD): mitochondrial dysfunction and neuroinflammation. However, the specific role of HMGB1 in PD remains elusive. We investigated the effect of 1-methyl-4-phenyl-1,2,3,6-tetrahydropyridine (MPTP) administration on mitochondrial dysfunction and HMGB1-associated inflammatory genes as well as locomotor activity in zebrafish, aiming to elucidate the role of HMGB1 in PD. Adult zebrafish received MPTP injections, and locomotor activity was measured at 24- and 48-h post-administration. Gene expression levels related to mitophagy (fis1, pink1, and park2) and HMGB1-mediated inflammation (hmgb1, tlr4, and nfkb) were quantified through RT-qPCR analysis. Following MPTP injection, the significant increase in transcript levels of fis1, pink1, and park2 indicated notable changes in PINK1/Parkin mitophagy, while the upregulation of hmgb1, tlr4, and nfkb genes pointed to the activation of the HMGB1/TLR4/NFκB inflammatory pathway. Furthermore, MPTP-injected zebrafish exhibited decreased locomotor activity, evident through reduced distance travelled, mean speed, and increased freezing durations. HMGB1 plays a major role in cellular processes as it is involved in both the mitophagy process and functions as a pro-inflammatory protein. MPTP administration in adult zebrafish activated mitophagy and inflammatory signaling, highlighting the significant role of HMGB1 as a mediator in both processes and further emphasizing its significant contribution to PD pathogenesis.

## 1. Introduction

### 1.1. Pathophysiology of PD

Parkinson’s disease (PD) is a neurodegenerative disorder marked by progressive dopaminergic neurodegeneration in the substantia nigra pars compacta (SNpc) [1]. After Alzheimer’s disease, PD is the second most common neurodegenerative disease, with global prevalence expected to double from 6.2 million cases in 2015 to 12.9 million cases by 2040 [2,3]. Several factors are associated with the risk of developing PD, including aging, genetic inheritance or mutation of PD-related genes, and environmental insults such as exposure to neurotoxins [4,5]. To date, the diagnosis of PD and the determination of disease progression rely on clinical manifestations and the severity of motor symptoms [6].

Although there are known risk factors and symptoms, the exact cause or origin of PD remains unknown. Nonetheless, accumulating evidence has revealed two factors that play a major role in PD pathogenesis: mitochondrial dysfunction [7,8] and neuroinflammation [1,9,10]. Investigating the link between these two factors is a continuing concern within PD pathogenesis.

### 1.2. High-Mobility Group Box 1 (HMGB1) in PD Pathogenesis

Dysregulations in mitochondrial genes, particularly those related to mitochondrial autophagy, also known as mitophagy, increases an individual’s susceptibility to developing PD [11]. The FIS1 (mitochondrial fission protein 1) participates in mitophagy by recruiting proteins involved in mitochondria-lysosome contacts [12]. The PINK1 (PTEN-induced kinase 1) accumulates on the outer membrane of damaged mitochondria and stimulates PARK2 (PRKN parkin RBR E3 ubiquitin protein ligase) to trigger selective autophagy via a signaling pathway known as the PINK1/Parkin mitophagy [13]. Mutations and perturbations in the expression of these genes have been reported in cellular [14,15] and animal [14,16,17,18] PD studies [14,15,16,17,18].

Researchers have elucidated the significant activation of the high-mobility group box 1 (HMGB1) protein in response to neuroinflammation [19,20,21]. During pathologic conditions, the HMGB1 protein acts as a damage-associated molecular pattern (DAMP) to modulate neuroinflammatory responses. Besides its potent role in the inflammatory process, HMGB1 plays a role in the development of neurodegeneration induced by mitochondrial dysfunction by stimulating both autophagy and apoptosis [19]. Moreover, HMGB1 is involved in the autophagic surveillance mechanism for maintaining mitochondrial quality control and works in concert with HSPB1 (heat shock protein beta 1) to regulate macro-autophagy and mitophagy [20]. Impaired mitochondrial function and increased neuroinflammation are believed to interact, triggering neurodegeneration [21]. However, despite the major influence of mitochondrial dysfunction and neuroinflammation on PD development, knowledge about the link between these two processes is still limited.

### 1.3. MPTP Neurotoxin Induces Mitochondrial Dysfunction in Animal PD Models

1-methyl-4-phenyl-1,2,3,6-tetrahydropyridine (MPTP) is a common neurotoxin used to induce PD in a variety of animal models ranging from primates to nematodes [22]. MPTP mimics dopaminergic degeneration by inhibiting the function of mitochondrial complex I in the electron transport chain (ETC), resulting in neuronal death [23]. The increasing popularity of zebrafish has widened their use as a model for neurodegenerative diseases. Published studies have investigated the effects of MPTP on adult zebrafish. These studies have demonstrated locomotor symptoms as well as molecular and histological changes following MPTP administration [22,24,25,26,27]. Previously, we conducted MPTP administration to adult zebrafish and monitored their swimming behavior for 96 h, revealing a pronounced impairment in locomotor activity starting at the 24th hour post-administration and persisting until the 96th hour [27]. Nevertheless, prior studies involving the administration of MPTP in adult zebrafish did not specifically explore the correlation between mitochondrial dysfunction and the inflammatory response triggered by MPTP neurotoxicity.

### 1.4. Relevance of Current Study

In this study, our main goal is to posit that the MPTP neurotoxin induces impairment of the mitophagy process, thereby triggering an HMGB1-mediated inflammatory response through the upregulation of gene expression related to these specific processes. Therefore, we administer MPTP to adult zebrafish via intraperitoneal (I/P) injection to investigate its effects on the expression levels of several mitochondria- and HMGB1-associated genes. Furthermore, to investigate the functional implications of the genetic findings, we aim to assess the locomotor activity of the MPTP-injected zebrafish. Based on our findings, MPTP is shown to increase the expression of mitochondrial *fis1*, *pink1*, and *park2* genes and HMGB1-mediated *tlr4* and *nfkb* genes. Moreover, we demonstrate that the observed decrease in locomotor activity in MPTP-injected zebrafish may potentially be attributed to the alterations in gene expression levels within this group, although these findings are preliminary. Hence, our study provides evidence that MPTP administration in adult zebrafish can establish a PD model with mitochondrial dysfunction and activate the HMGB1/TLR4/NFκB inflammatory signaling pathway. Although this study provides valuable insights into the genetic mechanisms underlying mitochondrial dysfunction-induced neuroinflammation in PD development, further investigations at the protein and molecular levels are necessary to enhance our understanding of the involved mechanisms and provide conclusive evidence.

## 2. Materials and Methods

### 2.1. Zebrafish Care and Husbandry

Adult zebrafish aged ~5 months old (0.8–1.0 g body weight) of the wild-type outbred short-fin strain were acquired commercially from a local supplier (Galing Aquarium, Kuantan, Pahang, Malaysia). All fish were experimentally naïve and acclimatized for at least two weeks in an 8 L home tank, filled with dechlorinated tap water maintained at 25–28 °C and equipped with a standard aquarium filter. Illumination was provided via a fluorescence light tube replicating a 14:10 h light/dark cycle (lights on and out at 8.00 a.m. and 10.00 p.m., respectively). All fish were fed at least once a day using commercial fish food micro-pellets (Sanyu Guppy, C.S.L. Thean Yeang Aquarium (M) Sdn. Bhd., Kuala Lumpur, Malaysia). After two weeks of acclimatization, actual experiments were performed. All procedures were approved by the Animal Care and Use Committee of the International Islamic University of Malaysia [ref. ID: IACUC2021-002(1)] and performed according to the standards of zebrafish care outlined by the Zebrafish Husbandry Association. The outbred model was used for this study due to its relevance and population validity. While it is true that inbred models offer genetic homogeneity and the potential for more reproducible results, it is important to note that modelling PD aims to simulate a genuine human condition that affects a genetically heterogeneous population. Therefore, incorporating genetically diverse models (outbred model) is crucial to capturing the complexity and variability of PD as it manifests in real human populations. Therefore, for the current study, using the outbred zebrafish model is deemed more translationally valid and relevant. Furthermore, both male and female zebrafish were selected in this study to better mimic the heterogeneity of human populations.

### 2.2. Preparation and Injection of MPTP Neurotoxin

Powdered 1-methyl-4-phenyl-1,2,3,6-tetrahydropyridine (C_12_H_15_N.HCl, Cat # M0896, Sigma-Aldrich, St. Louis, MO, USA) was dissolved in a saline solution (0.9% NaCl) to make 10 mg/mL stock. Fresh MPTP working solution was prepared on the day of the experiment by diluting the stock to 500 µg/mL. Prior to the administration, fish were anaesthetized in ice-cold water (10–12 °C). Once anaesthetized, 40 µL of MPTP solution was manually injected into the fish via the intraperitoneal (I/P) route using a 31 G needle attached to an insulin syringe. Control fish were anaesthetized and administered with 0.9% NaCl solution. Immediately after the injection, fish were placed in a recovery tank treated with anti-bacterial treatment (API^®^ MelaFix, Mars Fishcare North America Inc., Chalfont, PA, USA) and were monitored for recovery.

### 2.3. Brain Tissue Collection

The total brain tissue of fish injected with MPTP and controls were dissected carefully upon anesthesia in ice-cold water (2–4 °C). The collected brain tissues were rinsed twice in 1x PBS. Immediately, the tissues were immersed in RNAhold^®^ (TransGen Biotech Co., Ltd., Beijing, China, Cat# EH101-01) to inactivate RNase and keep RNA intact. The immersed tissues were incubated in 4 °C for 24 h, then stored in −20 °C until the next procedure.

### 2.4. Gene Expression Study

Total RNA from controls and MPTP-injected brain tissues were extracted using TRIzol reagent in accordance with the manufacturer’s protocol (TransGen Biotech Co., Ltd., Beijing, China, Cat# ER501-01). The purity and integrity of isolated RNA was determined using the NanoDrop™ One spectrophotometer (Thermo Fisher Scientific, Waltham, MA, USA) by measuring the ratio of absorbance at 260 nm and 280 nm. Samples only with absorbance of 1.8–2.1 were used for subsequent cDNA synthesis. 

First-strand cDNA was synthesized from the total RNA and subsequent qPCR reaction was performed using the TransScript^®^ Green Two-Step qRT-PCR SuperMix following the manufacturer’s protocol (TransGen Biotech Co., Ltd., Beijing, China, Cat# AQ201-01). All readings were analyzed using Bio-Rad CFX96 Touch Real-Time PCR Detection instrument (Bio-Rad Laboratories, Inc., Berkeley, CA, USA). The relative expression level (fold change) of the genes involved in mitochondrial dysfunction (*fis1*, *pink1*, and *park2*) and HMGB1-mediated inflammation (*hmgb1*, *tlr4*, and *nfκb*) was analyzed through the comparative Cq method using *actb* as the reference gene. The respective gene specific primer sequences are listed in Table 1.

### 2.5. Locomotor Assessment

The assessment of locomotor activity was conducted following the procedure outlined in our previous work [27], with necessary modifications made as required. The locomotor activity of all fish was assessed at baseline (before injection), 24th, and 48th hour post-injection. An assay tank sized 24.0 cm (L) × 13.5 cm (W) × 13.5 cm (H) filled with two liters of dechlorinated tap water was used for this assessment. The tank was digitally divided into four imaginary quadrants: Q1–Q4. The overview of the assay tank is given in Figure 1. The entire setup was walled with cardboards to minimize external auditory and visual noises, as well as experimenter effects. A light source was supplied by LED lights. A 16-megapixel mobile (Realme 3 Pro) camera with a frame rate of 30 fps was mounted on top of the assay tank to record the locomotor activity. The recorded videos were saved in an mp4 format.

Fish were placed individually in the tank and its locomotion was recorded for 3 min after a 5-min acclimatization period. Fish typically exhibit a natural behavior of swimming back and forth along the length of a tank. The locomotor activity was measured by recording the total distance travelled, mean speed, freezing duration, and crossing frequency during a 3-min observation. The total distance was defined as the distance that the fish moved during the 3 min session (in cm). The mean speed was calculated by dividing the total distance by 3 min (in cm/s). The freezing duration was defined as the cumulative duration of immobility throughout the 3 min (in sec). The crossing frequency refers to how many times a fish crosses from one quadrant to another within the 3 min. Upon completion of the 3-min recording period, fish were removed from the assay tank and the process repeated until all fish (both MPTP-injected and control groups) had been assessed and recorded.

The recorded videos were pre-processed to reduce noises and errors and were subsequently subjected to analysis using Noldus EthoVision XT version 11.5 tracking software. Velocity thresholds for adult zebrafish movement were established with stop and start velocities of 0.20 and 0.21 cm/s, respectively. Movements with velocities below 0.20 cm/s, such as slides, falls, sweeps, and freezing, were considered as non-swimming movements. The tracking images that were captured were saved in the form of .png files, while the resulting data from the analysis were exported as Excel files in the .xlsx format, which were then utilized for statistical analysis. Each parameter was compared between controls and MPTP-injected groups. The entire experimental procedure is outlined in a timeline format as illustrated in Figure 2.

### 2.6. Statistical Analysis

GraphPad Prism software version 9.0 (San Diego, CA, USA) was used to perform all statistical analyses. Gene expression study was analyzed using independent *t*-test. Locomotor activity parameters (total distance travelled, mean speed, freezing duration, and crossing frequency) were analyzed using repeated measures two-way ANOVA with post-hoc Sidak test as recommended by the statistical software. All results are presented as mean ± SEM, and *p*-values of less than 0.05 (*p* < 0.05) were considered to be statistically significant.

## 3. Results

### 3.1. Effect of MPTP on Mitochondrial and HMGB1-Associated Genes Expression

At the mechanistic level, MPTP causes PD-like mitochondrial dysfunction by inhibiting the activity of mitochondrial complex I. This inhibition leads to a reduction in ATP production and an increase in reactive oxygen species (ROS) population [30,31]. Damage or stress to the mitochondria can result in mitophagy [32]. Here, we examined the impact of MPTP administration on the expression of genes involved in mitophagy in adult zebrafish to provide further evidence of MPTP-induced mitochondrial dysfunction.

The global brain transcripts for three mitophagy-associated genes; *fis1*, *pink1*, and *park2*, were quantified using RT-qPCR. According to the analysis, the expression levels of *fis1* and *pink1* genes were found to have increased by three- and two-fold, respectively, in MPTP-injected zebrafish compared to the controls (Figure 3A). Additionally, the analysis revealed a significant increase in the *park2* gene, with its expression level reaching approximately seven times that of the controls (Figure 3A). Findings from the analysis suggest that the I/P delivery of MPTP can significantly upregulate the expression levels of genes involved in mitophagy and support that this method can be used to establish a model of PD in adult zebrafish.

Although neuroinflammation is a current focus of concern in PD, there is still a limited understanding of the impact of HMGB1-mediated neuroinflammatory response on the development of PD. Herein, we performed RT-qPCR analysis on the *hmgb1* gene of adult zebrafish to evaluate the effect of MPTP administration on the expression of HMGB1 transcript. The analysis revealed a significant 2.4-fold increase in the expression level of *hmgb1* gene of the MPTP-injected zebrafish relative to the controls (Figure 3B).

Furthermore, to investigate potential downstream neuroinflammatory pathways involved, we also examined the expression levels of two genes associated with HMGB1: *tlr4* and *nfkb*. The TLR4 has been identified as the main receptor for HMGB1 binding [33]. The mechanism underlying the HMGB1-dependent TLR4 signaling involves disulfide HMGB1 binding to MD-2 (myeloid differentiation factor 2) located on the cell surface of TLR4 [34]. Subsequently, the interactions between HMGB1 and TLR4 lead to the activation of NFκB. The RT-qPCR analysis revealed a 2.5-fold increase in *tlr4* and a 2-fold increase in *nfkb* gene expressions in MPTP-injected zebrafish compared to the controls (Figure 3B). Together, these data suggest that MPTP upregulates HMGB1 mRNA level and activates the HMGB1/TLR4/NFκB neuroinflammatory signaling pathway in adult zebrafish.

### 3.2. Effect of MPTP on Locomotor Activity of Adult Zebrafish

To investigate whether the observed gene expression changes translate into locomotor impairment, we monitored zebrafish swimming behavior before MPTP administration and at two time points following the administration (24th and 48th hour). The same fish were assessed throughout the trials to delineate individual variability within the group. 

We examined several key parameters that reflect locomotor activity, including total distance travelled, average speed, freezing durations, and frequency of crossing between quadrants. Initially, both groups of fish had similar locomotor activity prior to MPTP administration (Figure 4). In contrast, at the 24th hour after administration, MPTP-injected fish displayed nearly a 50% decrease in both total distance travelled and average speed compared to the controls. Further reductions were elicited at the 48th hour after administration (Figure 4A,B). Conversely, there was a significant increase in freezing durations, with a 66% increase observed at the 48th hour post-injection relative to the controls (Figure 4C). In addition, the frequency of crossing between quadrants decreased gradually with each trial point (Figure 4D). Representative path images visualize the track differences between the two groups, with the greatest disparity being observed at the 48th hour point (Figure 4E,F). In conclusion, the locomotor impairment induced by MPTP was evident 24 h after administration, and its severity gradually increased over the two-day locomotor assessment period.

## 4. Discussion

Recent literature provides substantial evidence regarding the impact of neuroinflammation on the development of PD, as evident from findings in both PD cases and animal models [35,36,37]. Significantly, there is a potential association between the neuroinflammatory processes observed in PD and mitochondrial dysfunction [19]. The HMGB1 protein, a pro-inflammatory nuclear protein that acts as DAMPs in pathologic conditions, plays a crucial role in maintaining mitochondrial homeostasis by regulating mitophagy [21]. Several reports have indicated that HMGB1 is overexpressed in the brain of PD patients and in animal models of PD [38,39,40]. Hence, the activation of HMGB1 may serve as a link that connects mitochondrial dysfunction to the neuroinflammatory mechanisms involved in the pathogenesis of PD.

In this study, we set out with the aim of examining the effect of MPTP on important genes related to mitophagy. Additionally, we investigated genes associated with HMGB1 to analyze the relationship between mitochondrial dysfunction and neuroinflammation in response to HMGB1 activation. We specifically focused on HMGB1 given its established involvement in maintaining mitochondrial quality control, particularly in the regulation of mitophagy [21,41], as well as the proven evidence of its overexpression in PD studies [38,39,40]. In this study, we administered a single dose of MPTP to adult zebrafish and assessed the effects through genetic analysis and locomotor assessment. The study findings demonstrated that MPTP administration resulted in increased expressions of mitochondrial genes *fis1*, *pink1*, and *park2*, as well as *hmgb1*, *tlr4*, and *nfkb* genes. Additionally, MPTP induced a gradual impairment in locomotor function in adult zebrafish.

### 4.1. Upregulation of Mitophagy-Associated Genes following MPTP Administration

In light of the crucial role of mitochondrial dysfunction in the neurodegenerative process of PD, our study aimed to investigate the impact of MPTP administration on genes associated with mitochondrial autophagy. Specifically, we conducted RT-qPCR analysis on the whole brain region of adult zebrafish to assess the expression levels of *fis1*, *pink1*, and *park2* genes in MPTP-injected fish compared to controls. Our results revealed that all three genes showed upregulation in the brains of MPTP-injected fish, consistent with the findings of Kalyn and Ekker [22], who examined the effects of cerebroventricular microinjection of MPTP in adult zebrafish. Similar observations were also reported by Sarath Babu et al. [25] regarding the expressions of *pink1* and *park2* genes.

The *fis1* gene encodes a protein known as FIS1, which plays a major role in the regulation of mitochondrial fission, a process by which mitochondria undergo fragmentation. In normal conditions, FIS1 serves as a receptor protein located on the outer membrane of mitochondria to facilitate the fission mechanism [12]. The fragmentation process is critical in mitochondrial quality control to maintain mitochondrial fusion-fission homeostasis [42]. Furthermore, FIS1 regulates the mitophagy process via recruiting other proteins involved in mitophagy [43]. Dysregulation of the *fis1* gene or its protein product has been evidenced to disrupt the normal process of mitophagy and disturb mitochondrial dynamics, resulting in mitochondrial dysfunction. Several studies have investigated the impact of mutations or alterations in the *fis1* gene on mitochondrial dynamics, revealing that such changes result in an excessive fragmentation of mitochondria and the accumulation of dysfunctional mitochondrial structures [44,45]. Imbalance in mitochondrial dynamics will, in turn, lead to ATP deficits and oxidative stress [46,47], both of which have been implicated in the pathogenesis of PD. In this regard, our finding of increased *fis1* gene expression following MPTP administration suggests a potential increase in the stimulation of mitochondrial fragmentation activity and/or the mitophagy process. While our study is limited to the genetic level, previous aforementioned studies provide supporting evidence for our hypothesis.

Besides FIS1, two other important proteins involved in mitochondrial quality control are the PINK1 protein kinase and Parkin E3 ubiquitin ligase, which are encoded by the *pink1* and *park2* gene, respectively. In particular, PINK1 plays a crucial part in identifying and damaged mitochondria, while Parkin is responsible for labelling and eliminating the mitochondria via mitophagy [13]. Therefore, PINK1 works closely with Parkin in the PINK1/Parkin mitophagy pathway to promote the clearance of damaged or dysfunctional mitochondria [48]. Mutations in *pink1* or *park2* genes have been described in PD studies [16,49,50], resulting in the disruption of the PINK1/Parkin signaling pathway. Impaired communication between these two proteins hinders the mitophagy process, leading to the accumulation of damaged mitochondria and the subsequent reduction in ATP production and increase in oxidative stress [51], which are key factors affected in PD. A study conducted on fruit flies has provided evidence that the loss of function of PINK1 or Parkin impairs the process of mitochondrial turnover and degradation [52]. Additionally, mutations in the *pink1* or *park2* genes have been identified as causative factors in the development of autosomal recessive early-onset PD [53]. In the context of our study, the increase in *pink1* and *park2* gene expression levels in the brains of MPTP-injected zebrafish suggest their involvement in neurotoxin-induced mitochondrial dysfunction.

The mechanism of MPTP action involves the inhibition of mitochondrial complex I activity, leading to reduced energy production and elevated oxidative stress [23]. Consequently, this disruption damages the mitochondria, promoting the initiation of the mitophagy process. As a result, mitophagy-associated proteins are recruited, and their gene expression levels are increased [54,55]. In line with this, our findings demonstrated that the administration of MPTP to adult zebrafish induced mitochondrial dysfunction, as evidenced by the upregulation of three key mitophagy-associated genes: *fis1*, *pink1*, and *park2*. These alterations in gene expression suggest an activation of the mitophagy process, emphasizing the profound impact of MPTP on disrupting mitochondrial function.

### 4.2. Upregulation of HMGB1-Mediated Inflammatory Genes following MPTP Administration

The next focus of our study concerned the effect of MPTP administration on HMGB1 gene expression. We discovered a notable elevation in HMGB1 mRNA levels in the brains of MPTP-injected zebrafish compared to controls. Our findings are consistent with the observations of Santoro et al. [56], who documented elevated HMGB1 levels in tissue and biofluid samples from PD patients, and with the study by Ren et al. [57], which reported a significant increase in HMGB1 mRNA levels in zebrafish larvae following MPTP administration, thus providing further support for our findings.

HMGB1 is a pro-inflammatory protein that plays a significant role in neuroinflammation [58,59], which has been recognized as a crucial factor in the development and progression of PD. Its overexpression and dysregulated release have been observed in the brains of PD patients and animal models [38,39,40], suggesting its involvement in PD pathogenesis. Studies investigating the serum and blood of individuals with PD have consistently shown elevated levels of HMGB1 protein [40,56]. Additionally, it has been observed that patients with poor responses to drug treatment exhibit higher levels of HMGB1 compared to patients with stable drug treatment outcomes [40]. Activation of HMGB1 can initiate downstream inflammatory cascades, leading to the progression of neuroinflammation in PD. HMGB1-mediated neuroinflammatory signaling triggers the release of inflammatory cytokines and chemokines and induces microglial activation [33]. Additionally, HMGB1 is involved in mitochondrial quality control mechanisms by cooperating with PINK1 and Parkin to regulate the process of mitophagy [21]. According to Zhang et al. [60], HMGB1 participates in the initiation of early-stage mitophagy while impairing the late-stage of mitophagy in microglia. Therefore, our findings suggest a plausible correlation between the upregulation of the *hmgb1* gene and the observed increase in mitophagy-associated genes, establishing a connection between mitochondrial dysfunction and HMGB1 activation in MPTP-injected adult zebrafish.

To further investigate the relationship between HMGB1-mediated inflammation and mitochondrial dysfunction in the context of MPTP neurotoxicity and to provide additional support for our findings, we evaluated the expression levels of two inflammatory genes associated with HMGB1, namely *tlr4* and *nfkb* genes. The observed increase in TLR4 mRNA expression in our study suggests the activation of TLR4 by HMGB1, thus initiating HMGB1/TLR4 inflammatory signaling in MPTP-injected zebrafish. Additionally, the observed upregulation of the *nfkb* gene can be attributed to HMGB1/TLR4 signaling, as it activates NFκB, a crucial transcription factor involved in the regulation of genes associated with inflammation. Activated NFκB protein subsequently translocates to the nucleus and binds to specific DNA sequences, leading to the transcription of pro-inflammatory cytokines and chemokines, such as TNF-α, IL-1β, and IL-6 [59], further exacerbating the neuroinflammatory response. The HMGB1/TLR4/NFκB signaling promotes the release of various pro-inflammatory cytokines and is implicated in the progression of PD [61]. Taken together, our results suggest that MPTP administration induces neuroinflammation in adult zebrafish through the HMGB1/TLR4/NFκB inflammatory pathway. 

Figure 5 depicts the interconnected pathways that link the genes investigated in our study, highlighting their relevant connections and interactions. Regarding HMGB1, it has a specific binding affinity for TLR4, as denoted by the purple arrow. Additionally, it exerts a direct impact on the expressions of TLR4 and NFκB, while being reciprocally influenced by the expression of PINK1, as illustrated by the blue arrows. HMGB1 directly regulates and is regulated by TLR4 and NFκB, establishing a causal-consequence loop between HMGB1, TLR4, and NFκB, highlighted by the grey arrows. Additionally, HMGB1 directly regulates the expression of PRKN (grey arrow) and is indirectly regulated by PRKN (dashed arrow).

In summary, our study demonstrated that MPTP administration led to the upregulation of *hmgb1*, *tlr4*, and *nfkb* genes in the brains of adult zebrafish. This suggests the involvement of HMGB1 in the inflammatory response following MPTP neurotoxicity. Additionally, the concurrent increase in mitophagy-associated genes indicates a potential role for HMGB1 in mediating PINK1/Parkin mitophagy signaling and triggering neuroinflammatory responses through the HMGB1/TLR4/NFκB pathway. Taken together, these findings emphasize the significant contribution of HMGB1 in the pathogenesis of PD, as it plays a role in both mitochondrial dysfunction and neuroinflammation, which are key factors in the development and progression of the disease.

### 4.3. Notable Locomotor Impairments following MPTP Administration

The compound 1-methyl-4-phenyl-1,2,3,6-tetrahydropyridine (MPTP) has been observed to negatively impact motor functions, as evidenced by a decrease in locomotor performance during swimming behavior tests. Consistent with previous studies [24,25], our results revealed that MPTP administration to adult zebrafish impaired locomotor function, indicated by slower swimming speed, shorter distance travelled, and longer freezing duration displayed by the MPTP-injected group. In addition, zebrafish injected with MPTP crossed the quadrants less frequently compared to the control group. These observations are consistent with the negative effects of MPTP on muscle control and coordination. MPTP, via its active metabolite MPP+, affects the dopaminergic system in the brain, resulting in a deficiency of the dopamine neurotransmitter, which is required for proper movement and motor control [62]. Dopamine depletion can cause slower movement and impaired motor function in animals injected with MPTP [63,64].

Parkinson’s disease is characterized by the disruption of motor functioning. In PD pathophysiology, the dopamine level decreases due to the progressive degeneration of dopaminergic neurons in the brain, particularly in the SNpc [1]. Dopamine deficiency disrupts the normal transmission of signals involved in movement control, causing motor symptoms such as tremors, rigidity, bradykinesia, and impaired coordination [65]. Similarly, MPTP neurotoxin can induce parkinsonian symptoms and mimic the pathologic mechanisms observed in PD. Toxic MPP+ accumulation hampers mitochondrial function, leading to impaired energy production (in the form of ATP) and increased ROS population [62]. As a consequence of this mitochondrial dysfunction, dopaminergic neurodegeneration ensues. Subsequently, the depletion of dopamine caused by the death of these neurons impairs motor control [66]. Since the key parameters in swimming behavioral tests are reliant on motor functioning, deterioration of the mechanisms involved in motor controls substantially impairs locomotor performance of MPTP-injected fish.

Several studies have documented the gradual impacts of MPTP on the nervous system. The gradual degeneration of dopaminergic neurons and the resultant deficiency in dopamine contribute to the progressive decline in the motor function of animals injected with MPTP. Markedly, the degree of severity and rapidity of MPTP effects can vary depending on factors such as the method of administration, dosage, and duration. A single dose of MPTP elicited notable locomotor impairment in adult zebrafish 24 h after administration, as observed in studies by Sarath Babu et al. [25], Selvaraj et al. [26], and Razali et al. [27]. Additionally, Razali et al. [27] reported gradual locomotor impairment in MPTP-injected zebrafish throughout their 96-h observation period. In a study by Kalyn and Ekker [22], gradual locomotor impairment was observed in adult zebrafish following multiple injections of MPTP via cerebroventricular. Moreover, the study showed that the most notable impact on locomotion was detected after the fourth injection. Consistent with previous findings, our results demonstrated a gradual disruption of locomotor function following MPTP administration, extending from the 24th hour to the 48th hour trial point. These results are consistent with those of previous studies, hence lending support to the notion that MPTP causes progressive locomotor impairment over time. Additionally, it is worth noting that the gradual impacts of MPTP reflect the progressive nature of dopaminergic neurodegeneration, mimicking the progressive development of PD. 

To briefly conclude, our results showed that MPTP administration to adult zebrafish impaired locomotor function, which was indicated by lesser distance travelled, slower swimming speed, and increased freezing durations. These findings suggest that MPTP induces parkinsonian motor symptoms by eliciting mitochondrial dysfunction and, consequently, activating HMGB1-mediated neuroinflammatory signaling in the adult zebrafish brain.

## 5. Conclusions

The available literature on PD neurodegeneration acknowledges the significant impact of mitochondrial dysfunction and neuroinflammation; nevertheless, it lacks a comprehensive understanding of the intricate relationship between disruptions in mitochondrial function and the mechanisms that trigger neuroinflammatory responses. In this study, we demonstrated that MPTP administration induces substantial expression levels of genes associated with the mitophagy process in the brains of adult zebrafish. Additionally, we observed a significant increase in HMGB1 mRNA expression and its associated downstream signaling genes in MPTP-injected zebrafish. The evidence from the gene expression analysis suggests that MPTP neurotoxicity affects mitochondrial function by activating the PINK1/Parkin mitophagy signaling pathway and triggering the HMGB1/TLR4/NFκB inflammatory response. Furthermore, the upregulation of these genes was reflected in the swimming behavior of the fish, as a decrease in locomotor activity was observed following MPTP administration. Moreover, we have developed a visual representation (Figure 5) illustrating the interplay among FIS1, PINK1, PRKN, HMGB1, TLR4, and NFκB. We propose that the targeted downregulation of *hmgb1* gene expression could serve as a potential adjunctive mechanism to mitigate impaired mitophagy and neuroinflammation in PD. Overall, our findings emphasize the significant role of HMGB1 in mediating mitophagy and triggering inflammatory responses, which are two crucial factors implicated in PD. This study contributes to our understanding of the relationship between mitochondrial dysfunction and inflammation induced by MPTP neurotoxicity and provides evidence for the involvement of HMGB1 in the pathogenesis of PD.

## Figures and Tables

**Figure 1 brainsci-13-01076-f001:**
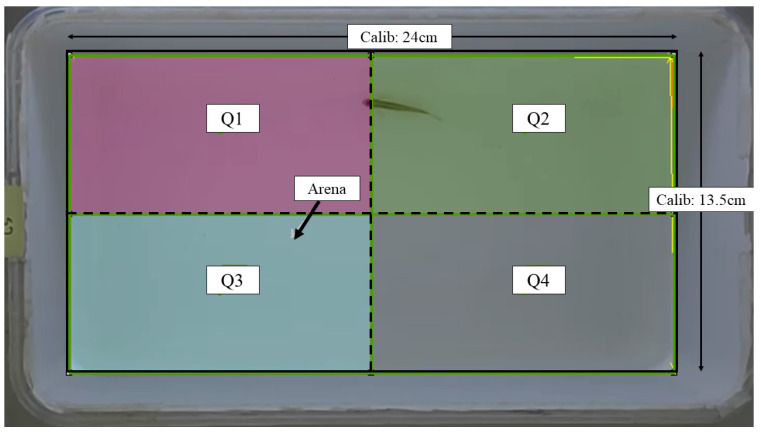
Schematic representation of the assay tank for locomotor assessment retrieved from Noldus EthoVision XT software version 11.5. Solid line: arena area; dashed line: imaginary quadrant divisions; Calib: calibration.

**Figure 2 brainsci-13-01076-f002:**
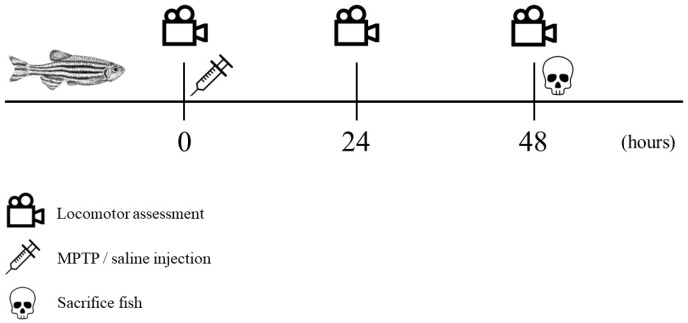
Chronological sequence of the experimental procedure. Locomotor assessment (indicated by a video icon) was conducted at three specific trial points: prior to injection, 24 h post-injection, and 48 h post-injection. Following the initial locomotor assessment, MPTP/saline was administered (indicated by a syringe icon). All fish were sacrificed immediately after the third locomotor assessment (indicated by a skull icon).

**Figure 3 brainsci-13-01076-f003:**
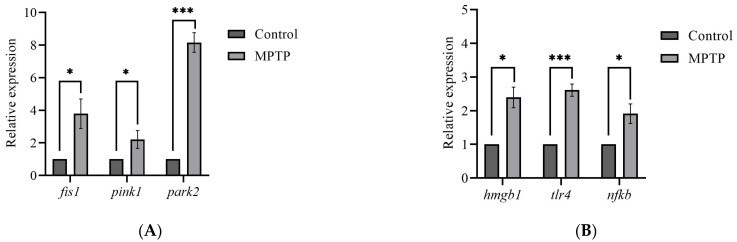
Effect of MPTP neurotoxin on mitochondria and HMGB1-associated genes in adult zebrafish. Whole RT-qPCR analysis for (**A**) mitochondrial genes; *fis1*, *pink1*, and *park2*, (**B**) HMGB1/neuroinflammatory genes; *hmgb1*, *tlr4*, and *nfkb* (*n* = 5 whole brains from each respective group). Bars represent the mean ± SEM. * *p* < 0.05, *** *p* < 0.005.

**Figure 4 brainsci-13-01076-f004:**
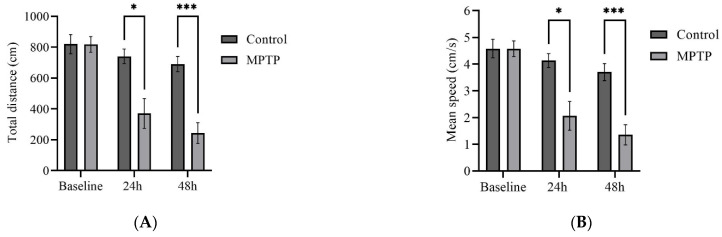
MPTP effect on adult zebrafish swimming behavior. (**A**–**D**) Analysis of total distance, average speed, freezing duration, and crossing frequency for MPTP-injected zebrafish and controls. Representative path images showing the movement of (**E**–**G**) controls, and (**H**–**J**) MPTP-injected zebrafish throughout the behavioral test period. Baseline, 24 h, and 48 h represent movement before MPTP/vehicle administration, 24 h, and 48 h after administration, respectively (*n* = 10 for both groups). Bars represent the mean ± SEM. * *p* < 0.05, ** *p* < 0.01, *** *p* < 0.005, **** *p* < 0.001.

**Figure 5 brainsci-13-01076-f005:**
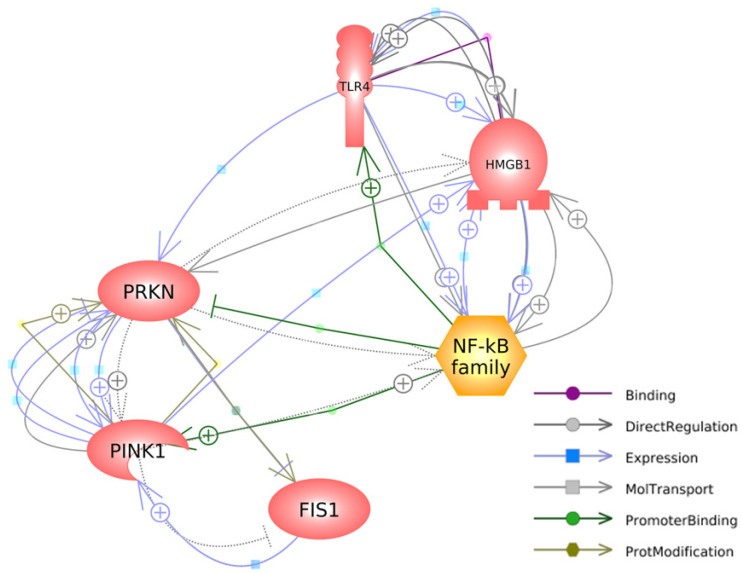
A gene network connecting mitophagy-associated and HMGB1-mediated inflammatory genes was generated with Pathway Studio^®^ (https://mammalcedfx.pathwaystudio.com/). Each link is supported by at least one published reference. Description on color-coded arrows can be found in the legends located in the bottom-right corner.

**Table 1 brainsci-13-01076-t001:** List of forward and reverse primer sequences for RT-qPCR.

Gene	Forward Sequence (5′-3′)	Reverse Sequence (5′-3′)	Ref.
*actb*	GGCATCACACCTTCTACAATGA	TACGACCAGAAGCGTACAGAGA	[25]
*park2*	ACAGACATCATGACTCCAGTGC	ACACGGAAATGATGAACCTCTT	
*pink1*	GGCAATGAAGATGATGTGGAAC	GGTCGGCAGGACATCAGGA	[22]
*fis1*	CCCTGAACCTTCCAGTGTTT	GTCTCTGGAAACGGGTCCTT	
*hmgb1*	GATGTGGCCTAGGTTCTTGTTC	CGTCGCATATACACAGCCTAAC	[28]
*tlr4*	TGCAAAGGCTTGTTCTTGTG	TGAAGGTGGTCATGAATGGA	[29]
*nfκb*	CGCAAGTCCTACCCACAAGT	ACCAGACTGTGAGCGTGAAG	

## Data Availability

The data presented in this study are available in this article.

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
