# Peer review of "Mitophagy: A Bridge Linking HMGB1 and Parkinson’s Disease Using Adult Zebrafish as a Model Organism"

_brainsci, 2023, doi:10.3390/brainsci13071076_

Round 1
Reviewer 1 Report
Comments and Suggestions for Authors
The writers have put up a piece that is extremely intriguing overall, however there are some improvements that need to be made.
The abstract should only be one paragraph long and should be formatted in the same way as structured abstracts, except it should not include headers.
In the introduction, you should quickly contextualize the research within a broader scope and emphasize the reasons why it is significant. It has to clarify the goal of the study and its relevance, including the hypotheses that are being tested specifically. It could be helpful to break the introduction into many subchapters.
The captions for Figure 2 need to provide a clearer explanation of the figure's content.
At the very least, a table need to be included.
More information needs to be included in the conclusion.
Author Response
Responses to Reviewer 1
We wish to thank Reviewer 1 for their insightful comments. Our point-to-point responses to the comments are shown below.
- The writers have put up a piece that is extremely intriguing overall, however there are some improvements that need to be made.
Response: We thank Reviewer 1 for their kind words.
- The abstract should only be one paragraph long and should be formatted in the same way as structured abstracts, except it should not include headers.
Response: We revised the abstract following MDPI Brain Sciences format. Accordingly, we discarded all headers (Lines 15-32).
- In the introduction, you should quickly contextualize the research within a broader scope and emphasize the reasons why it is significant. It has to clarify the goal of the study and its relevance, including the hypotheses that are being tested specifically. It could be helpful to break the introduction into many subchapters.
Response: We thank the Reviewer for this suggestion. Accordingly, we broke down the Introduction into four subchapters and added a sentence that specifically highlights the main goal (Lines 93-96), before briefly explaining the executive findings.
- The captions for Figure 2 need to provide a clearer explanation of the figure's content.
Response: We agree with the Reviewer and have made the necessary modifications by adding a more detailed description of Figure 2 (Lines 274-278).
- At the very least, a table need to be included.
Response: We encountered confusion regarding the table referred to by the Reviewer. Apart from Table 1, which presents the RT-qPCR primer sequences, we do not find any additional data that necessitates tabulation.
- More information needs to be included in the conclusion.
Response: We acknowledge this suggestion. We regret that our conclusion was somewhat inadequate. Accordingly, we added relevant sentences in the Conclusion (Lines 661-665). We hope there is now a more comprehensive portrait of the significance and findings of our study.
Reviewer 2 Report
Comments and Suggestions for Authors
Review of a manuscript “Mitophagy: A bridge linking HMGB1 and Parkinson’s Disease using Adult Zebrafish Model of Parkinson’s Disease” by Khairiah Razali and coauthors submitted to the “Brain Sciences”
Parkinson's disease is a severe disorder associated with dopaminergic neurodegeneration in the substantia nigra pars compacta. There is no efficient treatment modifying the course of this disease, so investigation of its genetics as well as molecular and cellular mechanism is important and timely. The authors provide a new insight into the genetic mechanisms underlying mitochondrial dysfunction and neuroinflammation in Parkinson's disease. This area of biomedical research is currently very important, and the data presented in the manuscript will be interesting for the readers of the “Brain Sciences”.
The following corrections and additions should be made.
Introduction
Lines 40-41 After the sentence “Several factors are associated with the risk of developing PD, including aging, genetic inheritance or mutation of PD-related genes, and environmental insults such as exposure to neurotoxins [4]” the authors should add a reference to a recent relevant review : ”Biomarkers in Parkinson’s Disease”. Chapter in a book Peplow et al., eds. Neurodegenerative Diseases Biomarkers. 2022. Neuromethods, 173, 155-180. Humana, NY. https://link.springer.com/protocol/10.1007/978-1-0716-1712-0_7
Lines 44-45: ”Despite the risk factors and symptoms, there is still no definite cause or origin that can explain the development of PD.”
The sense of this sentence is unclear, it should be rewritten in a easy to understand way.
Lines 55-56: ”Mutations and perturbations in the expression of these genes have been reported in various PD studies [13-17]. It should be clarified what type of studies the authors mean : human patients, animal or cell models, etc.
Lines 68-69: ”neuroinflammation on PD development, the link between these two processes is still limited.” Presumably the authors want to say that ”neuroinflammation on PD development, the knowledge about the link between these two processes is still limited.”
Line 88:” Furthermore, changes in these gene expressions translate into decreased fish locomotor activity”. How is this association proven?
Materials and Methods
Lines 110-112: ”Even though inbred model is more homogenous and can provide more reproducible results because of the genetic homogeneity, modelling PD involves real human condition affecting genetically heterogenous population.”
It is unclear what the authors mean by saying: ”modelling PD involves real human condition” Clarification needed.
Line 139. “Samples only with absorbance of 1.8-2.1 were used for subsequent cDNA synthesis.” Thre authors should add that the absorption at A260 was measured.
Line 148. “The respective gene specific primer sequences are listed in Table 1.” For some primers the references are given, for others – not. The authors should explain how they selected the sequence for primers without references.
Figure 1. Some of the fonts for the text on the figure are too small (calib 24 cm, arena 1) and should be increased for easy reading.
Figure 3 and 4 can be combined as Figure 3A and 3B.
Overall, interesting new data presented.
Author Response
Responses to Reviewer 2
We thank Reviewer 2 for their detailed and useful comments. We have addressed each of them as follows.
- Parkinson's disease is a severe disorder associated with dopaminergic neurodegeneration in the substantia nigra pars compacta. There is no efficient treatment modifying the course of this disease, so investigation of its genetics as well as molecular and cellular mechanism is important and timely. The authors provide a new insight into the genetic mechanisms underlying mitochondrial dysfunction and neuroinflammation in Parkinson's disease. This area of biomedical research is currently very important, and the data presented in the manuscript will be interesting for the readers of the “Brain Sciences”.
Response: We express our gratitude to the Reviewer for their kind words and for providing a summary of our study.
- Lines 40-41 After the sentence “Several factors are associated with the risk of developing PD, including aging, genetic inheritance or mutation of PD-related genes, and environmental insults such as exposure to neurotoxins [4]” the authors should add a reference to a recent relevant review: “Biomarkers in Parkinson’s Disease”. Chapter in a book Peplow et al., eds. Neurodegenerative Diseases Biomarkers. 2022. Neuromethods, 173, 155-180. Humana, NY. https://link.springer.com/protocol/10.1007/978-1-0716-1712-0_7
Response: We appreciate the Reviewer’s effort on providing additional reference for this study. We added the suggested reference and cited it accordingly in References (Lines 547-548).
- Lines 44-45: “Despite the risk factors and symptoms, there is still no definite cause or origin that can explain the development of PD.” The sense of this sentence is unclear, it should be rewritten in an easy-to-understand way.
Response: We agree with this suggestion. We revised the construct of the sentence for easier understanding (Lines 46-48).
- Lines 55-56: “Mutations and perturbations in the expression of these genes have been reported in various PD studies [13-17]. It should be clarified what type of studies the authors mean: human patients, animal, or cell models, etc.
Response: We appreciate this comment, and in response, we have clarified the sentence by specifying the type of studies as suggested. (Lines 59-61).
- Lines 68-69: “neuroinflammation on PD development, the link between these two processes is still limited.” Presumably the authors want to say that “neuroinflammation on PD development, the knowledge about the link between these two processes is still limited.”
Response: We thank the Reviewer for highlighting this concern. As per the suggestion, we have revised the sentence accordingly (Lines 73-75).
- Line 88:” Furthermore, changes in these gene expressions translate into decreased fish locomotor activity”. How is this association proven?
Response: In response to this comment, we have restructured the sentence to incorporate the finding regarding locomotor assessment. We showed that, while preliminary, the decrease in locomotor activity seen in MPTP-injected zebrafish may be the result of the changes in gene expression levels of this group (Lines 102-105).
- Lines 110-112: “Even though inbred model is more homogenous and can provide more reproducible results because of the genetic homogeneity, modelling PD involves real human condition affecting genetically heterogenous population.” It is unclear what the authors mean by saying: “modelling PD involves real human condition” Clarification needed.
Response: We agree with the Reviewer that the sentence is a bit confusing. What we meant was: the use of an outbred model in this study is particularly relevant due to its inherent heterogeneity, which closely mimics the genetic diversity observed in human populations. This choice allows for a more accurate representation of the diverse nature of PD in humans. We have restructured the sentence appropriately (Lines 126-131).
- Line 139. “Samples only with absorbance of 1.8-2.1 were used for subsequent cDNA synthesis.” The authors should add that the absorption at A260 was measured.
Response: We appreciate the comment and have revised the sentence accordingly by incorporating the measurement of absorbance as suggested (Line 160).
- Line 148. “The respective gene specific primer sequences are listed in Table 1.” For some primers the references are given, for others – not. The authors should explain how they selected the sequence for primers without references.
Response: We appreciate the Reviewer for bringing up this question. We apologize for the oversight in the previous version of the manuscript where some sequences shared the same references. We have rectified this issue by merging the table rows accordingly (Line 171).
- Figure 1. Some of the fonts for the text on the figure are too small (calib 24 cm, arena 1) and should be increased for easy reading. Figure 3 and 4 can be combined as Figure 3A and 3B.
Response: We agree with the comments. We changed to a new figure with clearer fonts (Line 184). Also, we combined Figures 3 and 4 as suggested and rearranged the structure of the Results section accordingly (225-284).
- Overall, interesting new data presented.
Response: We thank the Reviewer for their kind comments.